# Impact of Duodenal Stump Reinforcement in Preventing Duodenal Stump Fistula/Leakage After Distal or Total Gastrectomy for Malignant Disease: A Meta-Analysis of Comparative Studies

**DOI:** 10.3390/cancers17111735

**Published:** 2025-05-22

**Authors:** Maurizio Zizzo, Andrea Morini, Magda Zanelli, Giuseppe Broggi, Francesca Sanguedolce, Nektarios I. Koufopoulos, Andrea Palicelli, Lucia Mangone, Massimiliano Fabozzi, Mario Giuffrida, Candida Bonelli, Federico Marchesi

**Affiliations:** 1Surgical Oncology Unit, Azienda Unità Sanitaria Locale-IRCCS di Reggio Emilia, 42123 Reggio Emilia, Italy; andrea.morini@ausl.re.it (A.M.); massimiliano.fabozzi@ausl.re.it (M.F.); candida.bonelli@ausl.re.it (C.B.); 2Pathology Unit, Azienda Unità Sanitaria Locale-IRCCS di Reggio Emilia, 42123 Reggio Emilia, Italy; magda.zanelli@ausl.re.it (M.Z.); andrea.palicelli@ausl.re.it (A.P.); 3Department of Medical and Surgical Sciences and Advanced Technologies “G.F. Ingrassia”, Anatomic Pathology, University of Catania, 95123 Catania, Italy; giuseppe.broggi@phd.unict.it; 4Pathology Unit, Azienda Ospedaliero-Universitaria, Ospedali Riuniti di Foggia, 71122 Foggia, Italy; francesca.sanguedolce@unifg.it; 5Second Department of Pathology, Medical School, National and Kapodistrian University of Athens, Attikon University Hospital, 15772 Athens, Greece; nkoufo@med.uoa.gr; 6Epidemiology Unit, Azienda Unità Sanitaria Locale-IRCCS di Reggio Emilia, 42123 Reggio Emilia, Italy; lucia.mangone@ausl.re.it; 7Department of General Surgery, Azienda USL of Piacenza, 29121 Piacenza, Italy; m.giuffrida@ausl.pc.it; 8Clinica Chirurgica Generale, Azienda Ospedaliero-Universitaria di Parma, 43126 Parma, Italy; federico.marchesi@unipr.it

**Keywords:** duodenal stump, reinforcement, fistula, gastrectomy, gastric cancer, outcomes

## Abstract

The absence of duodenal stump reinforcement is considered one of the main risk factors for duodenal stump fistula (DSF) after distal/total gastrectomy for malignant gastric disease. Our meta-analysis of six comparative observational studies (19,527 patients: 11,545 reinforcement group versus 7982 control group) showed that, compared to the control group, the reinforcement group recorded a statistically significant lower DSF rate (OR: 0.32, 95% CI: 0.17, 0.60, *p* = 0.0004). Given the significant biases among meta-analyzed studies, our results require careful interpretation. Further randomized, possibly multicenter trials may turn out to be of paramount importance in confirming our results.

## 1. Introduction

According to the latest GLOBOCAN 2022 estimates produced by the International Agency for Research on Cancer (IARC) regarding the burden of cancer in the world, gastric cancer (GC) is the fifth most common cancer in the world with 968,350 new cases/year. Furthermore, it is the fifth leading cause of cancer-related mortality with 659,853 deaths/year [1].

As recommended by several recent international guidelines, standard gastrectomy is the main surgical procedure performed with curative intent [2,3,4,5]. It involves resection of at least two-thirds of the stomach with a D2 lymphadenectomy [2,3,4,5].

Among the different surgical procedures performed for GC, total gastrectomy and distal gastrectomy are among the most performed [3,5]. Total gastrectomy involves the total resection of the stomach including cardia and pylorus, while distal gastrectomy involves the resection of at least two-thirds of the stomach with preservation of the cardia [3]. Both procedures involve the creation of a duodenal stump at the end of the alimentary tract reconstruction.

There are several possible perioperative complications related to gastrectomy for GC with a wide range of incidence rates ranging from 11% to 46% [6]. They have been listed in detail in a recent international consensus [7]. One of the most feared is the duodenal stump fistula (DSF) (whose incidence ranges between 1.6% and 5%), which is associated with high rates of morbidity (75%) and mortality (16–20%), as highlighted in two Italian multicenter studies [8,9,10].

DSF was analyzed in several observational studies, mainly single-arm monocentric ones, in which a number of risk factors related to its occurrence were identified [9,10,11,12,13,14,15]. One of the most significant risk factors was the absence of duodenal stump reinforcement [8]. Current literature presents a few studies, also mainly single-arm observational ones, whose primary aim is to analyze the impact of duodenal stump reinforcement and the surgical method through which it is performed on the incidence of DSF [16,17,18,19,20,21,22]. The authors’ conclusion was that duodenal stump reinforcement was positive in reducing the DSF rate [16,17,18,19,20,21,22].

Our meta-analysis aimed to provide updated evidence by comparing DSF rates and other interesting short-term perioperative outcomes. The study involved patients who underwent distal or total gastrectomy for malignant gastric disease with or without reinforcement of the duodenal stump.

## 2. Materials and Methods

The present meta-analysis was performed following the Preferred Reporting Items for Systematic Reviews and Meta-Analyses (PRISMA) statement and guidelines [23]. As our meta-analysis was based on previously published studies and no addition of original patient population data was made, approval by an ethics committee and informed patient consent were not required. Our systematic review was not registered in a public registry.

### 2.1. Search Strategy

Potential articles of interest were identified through PubMed/MEDLINE, Scopus, Web of Science (Science and Social Science Citation Index), Embase, and Cochrane Library (Cochrane Database of Systematic Reviews, Cochrane Central Register of Controlled Trials (CENTRAL)) database consultation (16 March 2025).

After the first non-systematic search, a few results were identified in all five databases mentioned above. Therefore, we decided to reduce the number of keywords for the systematic search, omitting references to the minimally invasive approach and primary gastric pathology. The purpose of this was to avoid missing any potentially interesting articles.

The combination of non-MeSH/MeSH terms was as follows:PubMed/MEDLINE

((duodenal stump) AND (reinforcement)) AND (gastrectomy) Filters: English

((“duodenitis”[MeSH Terms] OR “duodenitis”[All Fields] OR “duodenum”[MeSH Terms] OR “duodenum”[All Fields] OR “duodenal”[All Fields]) AND (“amputation stumps”[MeSH Terms] OR (“amputation”[All Fields] AND “stumps”[All Fields]) OR “amputation stumps”[All Fields] OR “stump”[All Fields] OR “stumps”[All Fields] OR “stump s”[All Fields]) AND (“reinforce”[All Fields] OR “reinforced”[All Fields] OR “reinforcement, psychology”[MeSH Terms] OR (“reinforcement”[All Fields] AND “psychology”[All Fields]) OR “psychology reinforcement”[All Fields] OR “reinforcement”[All Fields] OR “reinforcements”[All Fields] OR “reinforcer”[All Fields] OR “reinforcer s”[All Fields] OR “reinforcers”[All Fields] OR “reinforces”[All Fields] OR “reinforcing”[All Fields]) AND (“gastrectomy”[MeSH Terms] OR “gastrectomy”[All Fields] OR “gastrectomies”[All Fields])) AND (english[Filter])

Scopus

(ALL (duodenal AND stump) AND ALL (reinforcement) AND ALL (gastrectomy)) AND (LIMIT-TO (LANGUAGE, “English”))

Web of Science

duodenal stump (Topic) AND reinforcement (Topic) AND gastrectomy (Topic) and English (Languages)

Embase

duodenal AND stump AND reinforcement AND gastrectomy AND [english]/lim

Cochrane Library

Duodenal stump in All text AND reinforcement in All text AND gastrectomy in All text—(Word variations have been searched) Language: English

Final search was carried out on 16 March 2025.

Moreover, the reference lists of included studies and relevant reviews were manually searched.

### 2.2. Inclusion Criteria

This study included only comparative population studies such as case series, case–control studies, cohort studies, controlled clinical trials, and randomized clinical trials (RCTs) for adult patients (≥18 years old) undergoing distal/total gastrectomy with or without reinforcement of the duodenal stump for malignant gastric disease.

In detail, we present our inclusions in compliance with the PICOS criteria:

Population (P): adult patient populations undergoing distal/total gastrectomy for malignant gastric disease;

Intervention (I): distal/total gastrectomy with reinforcement of duodenal stump;

Comparators (C): distal/total gastrectomy without reinforcement of duodenal stump;

Outcomes (O): duodenal stump fistula/leakage rate;

Study designs (S): comparative RCTs and non-RCTs.

Abstracts, posters, letters to the Editor, editorials, case reports, and previously published systematic reviews and/or meta-analyses were ruled out.

Due to poor data retrieved during the first unsystematic search, our systematic search ruled out restrictions as far as date of issue was concerned.

### 2.3. Outcomes

We evaluated two groups of outcomes: primary and secondary ones.

Primary outcome: DSF rate.

Secondary outcomes: operative time; estimated blood loss (EBL), overall postoperative complication rate; major postoperative complication (Clavien–Dindo ≥ 3 or CD ≥ 3) rate; length of hospitalization.

### 2.4. Data Extraction

Papers were selected and identified by two independent reviewers (M.Zi. and A.M.) based on title, abstracts, keywords, and full texts. The screening of the manuscripts was performed through the website Rayyan.ai (Rayyan Systems, Inc., Cambridge, MA, USA) [24].

The discrepancies arising from the selection process were overcome through discussion between the two reviewers until a consensus was reached. To increase the accuracy of the selection process, a double-blind method was followed which led to a high and satisfactory inter-observer agreement (Kappa = 0.92).

The following data were collected from included papers:Demographic data (author’s surname and year of publication, study type, study centers, study country, study period, population size, size of population with duodenal stump reinforcement, DSF rate, gender and age, body mass index (BMI), American Society of Anesthesiologists (ASA) score, neoadjuvant chemotherapy ± radiotherapy);Surgical data (surgical approach, type of gastrectomy and lymph node dissection);Pathological data (pT, pN, stage of disease);Duodenal stump reinforcement data (stapler type, cartridge length and closure height, reinforcement method, suture thread type).

All the results collected were eventually examined by a third independent reviewer (M.F.).

### 2.5. Quality Assessment

Two independent reviewers evaluated the quality of the included comparative studies using Version 2 of the Cochrane Risk of Bias tool (RoB 2) for randomized trials and the Risk Of Bias In Non-randomized Studies - of Interventions, Version 2 (ROBINS-I V2) for non-randomized ones [25,26].

RoB 2 was used for assessing the risk of bias in randomized trials [25]. A fixed set of bias domains focusing on different aspects of study design, conduct, and reporting are included in the tool [25]. Each domain had a series of questions (“reporting questions”) aimed at collecting data on study features [25]. A proposal for bias risk from each domain was generated by an algorithm, based on answers to reporting questions [25]. Ratings for risk of bias were “Low”, “High”, or “Some Concerns” [25].

The ROBINS-I V2 tool was used to assess the risk of bias in a specific result from an individual non-randomized study that examines the effect of an intervention on an outcome [26]. It compared health outcomes of two or more interventions [26]. To obtain an assessment of the risk, reporting questions were used that had a substantial factual nature and aimed at easing judgment on the risk of bias [26]. Answers to the reporting questions provided a framework for domain-level judgments on the risk of bias, which then served as a basis for an overall evaluation on the risk of bias in a special outcome [26]. Ratings for risk of bias judgments were “Low Risk”, “Moderate Risk”, “Severe Risk” and “Critical Risk”, keeping in mind that “Low risk” meant the risk of bias in a high-quality randomized study [26].

### 2.6. Statistical Analysis

Our meta-analysis was performed using Review Manager (RevMan) Version 5.4 (the Cochrane Collaboration, 2020) [27]. For dichotomous outcomes, odds ratios (ORs) and corresponding 95% confidence intervals (CIs) were computed according to Mantel–Haenszel (MH) method. For continuous outcomes, weighted mean differences (WMDs) and corresponding 95% CIs were computed by use of inverse variance (IV) method.

In the presence of an endpoint with median and range or median and interquartile range (IQR), the mean and standard deviation (SD) were calculated using the Wan formulae [28]. Instead, the Cochrane formula was adopted to combine the means and SDs of two or more groups into a single group having individual mean and SD [29].

I^2^ statistics were used to assess statistical heterogeneity. Here, <25, 25–50, and >50% I^2^ values were classified as follows: low, moderate, and high. Given the discrepancies between the included populations in terms of general population characteristics, surgical procedures, and histopathology of the surgically treated primary lesions, a random-effects model was used as default in all statistical analyses. Statistical significance was set at *p* < 0.05.

Sensitivity analysis was performed using the leave-one-out method to identify the influence of each study on the overall effect-size estimate and influential studies. In addition, subgroup analyses were performed to evaluate the impact on the outcomes of interest of both the specific surgical method adopted for duodenal stump reinforcement and duodenal stump reinforcement alone in the minimally invasive group.

Egger’s test was used to assess the publication bias.

## 3. Results

### 3.1. Search Results

The final systematic search performed in March 2025 identified 121 references (Figure 1). Following the removal of duplicate or irrelevant articles by title and abstract, the remaining full-text articles were assessed for inclusion. Out of these, six were comparative studies on the topic of interest and, therefore, were subjected to qualitative and quantitative analysis [16,19,21,22,30,31].

### 3.2. Quality of Studies

According to ROBINS-I V2, four non-randomized studies revealed moderate overall bias [19,22,30,31] and two showed serious overall bias [16,21] (see Appendix A). The RoB 2 tool was not employed due to lack of identification of randomized trials.

### 3.3. Study and Population Characteristics

Table 1, Table 2 and Table 3 show the general, surgical, and pathological characteristics of the included populations. The six included studies were all observational ones with a retrospective design and came exclusively from Asian countries [16,19,21,22,30,31]. The overall study period covered approximately 20 years (2005–2023) with a pooled population of 19,527 patients analyzed [16,19,21,22,30,31]. Males represent 70% of the pooled population (13,684/19,527) with a mean age between 59.81 and 69.22 years [16,19,21,22,30,31]. Just over half of the pooled population underwent minimally invasive surgery (53.67%; 10,481/19,527) and almost identical distributions of patients between distal gastrectomy (50.86%; 9932/19,527) and total gastrectomy (49.14%; 9595/19,527) [16,19,21,22,30,31] as well as between D2 dissection (52.1%; 9200/17,663) and D1+ dissection (47.9%; 8463/17,663) were identified [16,19,21,31]. The overall DSF rate is 1.02% (199/19,527) with a rate of 1.39% in the population without duodenal stump reinforcement and 0.76% in the population undergoing duodenal stump reinforcement [16,19,21,22,30,31]. The definitions of DSF were not always provided [16,19]. Additionally, the definitions presented by four of the six included studies exhibited differences (see Appendix A) [21,22,30,31].

Many other interesting specific features are available in Table 2 and Table 3.

### 3.4. Duodenal Stump Reinforcement Methods

From the analysis of surgical methods of duodenal stump reinforcement described in the comparative studies included in this meta-analysis, the presence of different approaches emerged, which we detail in Table 4.

### 3.5. Meta-Analyses Results

#### 3.5.1. Duodenal Stump Fistula/Leakage

Comparison of DSF rates in patients with or without duodenal stump reinforcement was analyzed in all six included comparative studies (Figure 2) [16,19,21,22,30,31]. Meta-analysis showed a statistically significant reduced DSF rate in the duodenal stump reinforcement group (OR: 0.32, 95% CI: 0.17, 0.60, *p* = 0.0004). The identified heterogeneity was moderate but not statistically significant (I^2^ = 48%, *p* = 0.09).

#### 3.5.2. Operative Time

Five out of six included comparative studies showed a comparison of the operative time between the two groups (Figure 3) [16,19,21,22,31]. The meta-analysis of the pooled population highlighted the absence of statistically significant differences between the two groups (MD: 15.31, 95% CI: −16.97, 47.59, *p* = 0.35). However, heterogeneity was high and statistically significant (I^2^ = 97%, *p* < 0.00001).

#### 3.5.3. Estimated Blood Loss

EBL was analyzed in five of the six included comparative studies (Figure 4) [16,19,21,22,31]. Meta-analysis of the pooled population showed no statistically significant differences between the two groups (MD: 8.86, 95% CI: −23.90, 41.61, *p* = 0.60). Also, for the above-mentioned outcome the heterogeneity was high and statistically significant (I^2^ = 89%, *p* < 0.00001).

#### 3.5.4. Overall Postoperative Complications

Four of the six included comparative studies analyzed the overall postoperative complication rate (Figure 5) [16,21,22,31]. There was no statistically significant difference between the two groups in the meta-analysis of the pooled population (OR: 0.87, 95% CI: 0.63, 1.19, *p* = 0.38). Heterogeneity was low and without statistical significance (I^2^ = 0%, *p* = 0.62).

#### 3.5.5. Major Postoperative Complications (Clavien–Dindo or CD ≥ III)

Five out of six included comparative studies presented a comparison of the major postoperative complications between the two groups (Figure 6) [16,19,21,22,31]. The meta-analysis of the pooled population highlighted the presence of statistically significant differences between the two groups, in favor of the duodenal stump reinforcement group (OR: 0.66, 95% CI: 0.43, 0.99, *p* = 0.04). Heterogeneity was low and without statistical significance (I^2^ = 0%, *p* = 0.59).

#### 3.5.6. Length of Hospital Stay

The length of hospital stay was analyzed in just three of the six included comparative studies (Figure 7) [16,21,22]. Meta-analysis of the pooled population showed no statistically significant differences between the two groups (MD: −0.45, 95% CI: −1.65, 0.75, *p* = 0.46). However, the heterogeneity identified was high and statistically significant (I^2^ = 87%, *p* = 0.0005).

#### 3.5.7. Sensitivity and Subgroup Analyses

Leave-one-out sensitivity analysis confirmed the primary and secondary outcomes of the pooled population after exclusion of the Gu et al. [30], Wang et al. [22], and Sano et al. [31] studies. However, major postoperative complications lost statistical significance after exclusion of the Inoue et al. [16], Ri et al. [19], and Sun et al. [21] studies (see Appendix A).

A first subgroup analysis was carried out because of discrepancies in the duodenal stump reinforcement method adopted. It confirmed the primary outcome of pooled analysis (see Appendix A). In particular, both duodenal stump reinforcement with seromuscular suture of any type (OR: 0.32, 95% CI: 0.18, 0.58, *p* = 0.0001; I^2^ = 40%, *p* = 0.14) and duodenal stump reinforcement with seromuscular interrupted suture (OR: 0.19, 95% CI: 0.07, 0.49, *p* = 0.0007; I^2^ = 0%, *p* = 0.38) showed a statistically significant reduced rate of DSF compared to the non-reinforcement group.

A second subgroup analysis was carried out because of discrepancies in the surgical approach adopted. This analysis also confirmed the primary outcome of pooled analysis (see Appendix A). Just considering the pooled population undergoing laparoscopic gastrectomy, duodenal stump reinforcement alone (OR: 0.23, 95% CI: 0.13, 0.43, *p* < 0.00001; I^2^ = 0%, *p* = 0.66), duodenal stump reinforcement with seromuscular suture of any type (OR: 0.23, 95% CI: 0.13, 0.43, *p* < 0.00001; I^2^ = 0%, *p* = 0.66), and duodenal stump reinforcement with seromuscular interrupted suture (OR: 0.19, 95% CI: 0.07, 0.49, *p* = 0.0007; I^2^ = 0%, *p* = 0.38) showed a statistically significant reduced rate of DSF compared to the non-reinforcement group.

#### 3.5.8. Publication Bias

According to the Cochrane Handbook for Systematic Reviews of Interventions (Version 5.1.0), tests for funnel plot asymmetry should be performed just in meta-analyses of at least 10 studies [32]. As our meta-analysis included six studies, we did not carry out analysis of publication bias. Indeed, fewer studies reduce the power of tests to identify the case from real asymmetry [32].

## 4. Discussion

To the best of our knowledge, our meta-analysis is the first to analyze the impact of duodenal stump reinforcement performed in distal or total gastrectomy for malignant gastric disease exclusively based on comparative studies.

We identified six comparative studies with a pooled population of 19,527 patients, slightly more than half of whom (59%) underwent duodenal stump reinforcement [16,19,21,22,30,31]. The primary outcome of our meta-analysis was the rate of DSF, which was significantly reduced in the reinforcement group compared to the non-reinforcement group with an OR of 0.32 (*p* = 0.0004).

The significant correlation between the reinforcement of the duodenal stump and the rate of DSF identified in the present meta-analysis seems to confirm what has already been presented in the literature. Over the last two decades, several authors have examined the potential risk factors related to DSF after GC surgery through univariate and multivariate analyses performed in observational studies [8,9,10,11,12,30,33,34]. They can be divided into three macro groups: risk factors related to (i) patient characteristics, (ii) primary gastric cancer-related conditions, (iii) intraoperative procedures [8]. Among all those presented, the absence of duodenal stump reinforcement was one of the most significant independent risk factors for DSF [8].

Not all included studies reported an analysis to identify risk factors for DSF [16,21,24]. In the univariate analyses, several risk factors were identified: age [22], sex [19], BMI [19,22,30], ASA score [22], EBL > 30 mL [19], preoperative C-reactive protein [30], preoperative albumin [30], tumor size [22], T stage [30], and lack of duodenal stump reinforcement [19,30]. Age [22], BMI [19,22,30], ASA score [22], preoperative C-reactive protein [30], and lack of duodenal stump reinforcement [19,30] were identified as significant independent risk factors for the development of DSF in multivariate analyses, adjusted for potential confounding factors.

Recently, Li et al. presented an interesting study aiming to establish a machine learning-based predictive model to estimate the occurrence of DSF in patients undergoing laparoscopic gastrectomy for GC [35]. Using data from a population of 4070 patients, specifically incorporating 11 clinical–pathological features to build machine learning models, the authors demonstrated that the support vector machine (SVM) model independently predicted DSF in GC patients and showed favorable discrimination and accuracy [35]. Based on this result, an efficient and intuitive online predictive tool has been constructed with significant potential in the prevention of DSF [35]. The above-mentioned model identified tumor site and stage, operative time, preoperative pyloric obstruction, patient age, and duodenal stump reinforcement as factors with significant impact on the occurrence of DSF after surgery for GC [35].

An interesting aspect to consider was the duodenal reinforcement method adopted. The authors of the studies included in our meta-analysis described different methods: seromuscular linear interrupted sutures [16,19,21,30], seromuscular linear continuous sutures [22,30], seromuscular purse-string sutures [21,22,30], and reinforced staplers [31]. We attempted to perform subgroup analyses aimed at analyzing the impact of individual reinforcement methods on DSF rate. Unfortunately, we were able to perform just two subgroup meta-analyses, regarding seromuscular sutures of any type (excluding patients treated with reinforced staplers) and those specifically with seromuscular linear interrupted sutures. Both methods demonstrated a statistically significant DSF rate reduction in the reinforcement group compared to the non-reinforcement group, with ORs of 0.32 (*p* = 0.0001) and 0.19 (*p* = 0.0007), respectively. There was no possibility of performing subgroup analyses comparing two reinforcement methods in the absence of sufficient available data.

The only two subgroup analyses that it was possible to perform demonstrated that seromuscular reinforcement significantly impacts DSF rate regardless of how it is performed. These results appear to be consistent with current literature. In fact, several single-arm observational studies demonstrated the potential benefit of different seromuscular reinforcement methods (barbed suture method, buried suture method, handover method) [17,18,20,36,37,38]. Other duodenal stump reinforcement methods such as bioabsorbable polyglycolic acid (BPA) felt in combination or not with fibrin glue or a reinforced linear stapler with PGA sheets appeared to be effective in DSF rate reduction [12,39].

In addition to the duodenal stump reinforcement method adopted, it was interesting to analyze how relevant the reinforcement itself can be during a laparoscopic approach. Duodenal stump reinforcement with seromuscular sutures is often not performed during laparoscopic distal/total gastrectomy due to technical difficulties [10,31,40]. Previously published observational studies highlighted a significantly higher rate of DSF after laparoscopic gastrectomy compared to open gastrectomy [11,41]. Therefore, it was assumed that the main reason related to this result was the omission of duodenal reinforcement, as underlined by Sano et al. [31].

In this regard, we performed a subgroup analysis related to comparative studies including the laparoscopic approach only. Unfortunately, we had to exclude the one by Sano et al., representing 84.4% of the pooled population (16,475 patients). This was due to the presence of patients undergoing open (9046 patients) and robotic (873 patients) gastrectomy and without the availability of individual-patient data to be able to meta-analyze. In both our analyses, seromuscular reinforcement of any type and that with interrupted suture, duodenal stump reinforcement appeared to be related to a significant reduction in the rate of DSF (OR: 0.23, *p* < 0.00001 and OR: 0.19, *p* = 0.0007, respectively).

The absence of individual-patient data in the study with the largest sample size significantly reduced the statistical power of the subgroup results. The creation of national registries and the enhancement of existing ones (e.g., Dutch Upper Gastrointestinal Cancer Audit—DUCA [42]) could significantly support the scientific community. The diverse nature of registry data (epidemiological, clinical, laboratory, surgical, histopathological, biomolecular, health system and organizational level, costs, etc.) would enable us to gather a wealth of information that can be linked to individual national contexts and specific national subpopulations. Subsequently, collaboration between the centers through the development of well-designed multicenter studies would facilitate the achievement of results that could significantly impact clinical practice. The aforementioned can be fully extended to multinational/regional entities, as seen with the existing National Cancer Database in the United States [43] and GASTRODATA in Europe [44]. However, they too require significant enhancement, as they lack several crucial parameters.

In the absence of RCTs, which can be challenging to construct, designing multicenter studies from registries offers numerous advantages [45,46]. These include rapid recruitment of a large number of participants, documentation of population and subpopulation diversity, statistical power, improved external validity and generalizability, greater relevance and practical application possibilities, increased funding appeal, networking, and enhanced academic and peer recognition [45,46]. Last but not least, it would enable the participation of low-volume centers or those unable to support a single-center study due to economic, organizational, or healthcare system constraints.

Our results would seem to confirm that the most effective method to reduce the risk of DSF is duodenal transection with a linear stapler and subsequent reinforcement of the staple line.

However, several factors could have influenced the short-term results obtained by the authors. The analysis of the included studies revealed that they were exclusively from Eastern countries, and there was a significant lack of data on characteristics that could potentially impact short-term outcomes [16,19,21,22,30,31]. The extensive literature on GC surgery provided numerous risk prediction models for perioperative outcomes and survival after gastrectomy for GC [47,48,49,50,51]. These risk prediction models, with or without machine learning methods, identified several important variables that contribute to morbidity and mortality following gastrectomy: age, BMI, Eastern Cooperative Oncology Group (ECOG) score, ASA score, history of severe pulmonary or cardiac disease, preoperative albumin, preoperative hemoglobin, type of surgery, and others [48,49,50,51]. Other multicenter studies identified additional risk factors through multivariate analysis, such as neoadjuvant chemotherapy [52,53]. Additionally, it is important to consider the significant differences between geographical regions [54,55]. Western GC patients tend to be older, have a higher BMI, more comorbidities, more locally advanced GCs, and undergo more neoadjuvant chemotherapy treatments compared to Eastern GC patients [56,57,58]. In contrast, Eastern countries frequently opt for laparoscopic surgery and D1 lymphadenectomy, based on early GC diagnosis through screening [56,57,58]. They have more minimally invasive GC surgery experience and a larger hospital volume [56,57,58]. Considering the details mentioned above, the exclusive focus on Eastern studies and their heterogeneity in the characteristics of the included populations hinder the external validation of our results, particularly for Western populations.

Furthermore, the potential impact of the center volume and the experience of the first surgeon would seem relevant. Two recent meta-analyses explored the correlation between hospital volume and perioperative outcomes [59,60]. In the study by Ji et al., 53 studies on the impact of hospital (48) or surgical (11) volume on 11 outcomes were identified [59]. The quantitative analysis revealed that gastrectomies for GC in high-volume centers were associated with lower short-term mortality, shorter hospital stays, and improved overall survival [59]. A higher surgeon volume was associated with a lower 30-day mortality rate [59]. However, the authors emphasized that the question of whether the hospital volume or the surgeon volume was more significant remained unanswered [59]. Similarly, Ning et al. found that the risk of postoperative mortality was 35% lower in patients undergoing cancer-related gastrectomy at high-volume centers [60]. The volume–outcome analysis revealed that this risk remained stable or decreased after the hospital volume reached a plateau of 100 gastrectomies per year (from 17.7% to 0.3%, with a stable trend below 0.51%) [60]. Among the included studies, only Sano et al. explored the impact of hospital volume [31]. The multicenter KSCC DELICATE study by Sano et al. demonstrated that the incidence of DSF was significantly lower in high-volume institutions [31]. Furthermore, a detailed comparison between high- and low-volume institutions showed that high-volume institutions had a significantly higher rate of duodenal stump reinforcement, despite various clinical–pathological differences [31]. These results suggested the importance of duodenal stump reinforcement with seromuscolar suture to prevent DSL.

Several published observational studies explored the learning curve of surgeons in minimally invasive gastrectomy for GC [61,62]. To date, there are only two meta-analyses on the learning curve of distal or total gastrectomies for GC [61,62]. Chan et al. demonstrated that approximately 44 and 21 cases were required to overcome the learning curve in laparoscopic and robotic total gastrectomies, respectively [24]. In contrast, the authors themselves identified approximately 47 and 22 cases required to overcome the learning curve in laparoscopic and robotic distal gastrectomies [23]. However, the studies included in the meta-analyses had several limitations that affected the significance of the results: (i) different outcome parameters (e.g., operative time, EBL, overall or major complications); (ii) non-arbitrary versus arbitrary cut-offs; (iii) single-surgeon learning curve versus institutional learning curve; (iv) minimally invasive assisted versus totally minimally invasive gastrectomy; (v) intracorporeal versus extracorporeal reconstructions; (vi) different reconstruction methods. None of the studies included in our meta-analysis explored the impact of surgeon experience on perioperative outcomes [16,19,21,22,30,31].

Another aspect to consider is the impact of the increasingly widespread robotic surgery [63]. It could guarantee a simpler, safer, and more effective technical execution of the seromuscular duodenal reinforcement through the various advantages that characterize and differentiate it from laparoscopic surgery (three-dimensional imaging, tremor filter, improved dexterity with an internal articulated EndoWrist—Intuitive Surgical Inc., Sunnyvale, CA, USA—that allows seven degrees of freedom), with a potential impact on the DSF rate [63]. Sano et al. were the only ones to analyze patients undergoing robotic gastrectomy in the pooled population [31]. However, the absence of individual-patient data did not allow analysis of the correlation between duodenal stump reinforcement and DSF in this specific subgroup [31]. Nor does the DSF rate seem to be one of the main topics of interest in the comparison of laparoscopic gastrectomy/robotic gastrectomy, as evident from the latest published meta-analyses [64,65,66,67].

Finally, the use or not of techniques for evaluating the correct visceral vascularization such as the indocyanine green (ICG) test could have influenced the results obtained. However, nothing was reported in this regard in the included studies.

For all secondary outcomes, we can only speculate on the factors that may have influenced the results. The operative time is potentially influenced by patient-related (high BMI, previous abdominal surgery history), primary surgical pathology-related (locally advanced GC), and surgical procedure-related (laparoscopic approach, total gastrectomy, D2 lymphadenectomy) factors [68,69,70,71,72,73,74,75]. Additionally, the surgeon’s experience plays a significant role. Ri et al. [19] and Sun et al. [21] were the only ones to demonstrate statistically significant differences between the two groups, with the non-reinforcement group showing a clear advantage. Both hypothesized that the laparoscopic reinforcement technique was the reason for the extended operative times [19,21].

EBL shares many of the factors mentioned above with the operative time, except for the surgical approach. Open gastrectomy is significantly linked to EBL [70,71,72]. Sano et al. demonstrated a significant statistical advantage for the non-reinforcement group, with the high rate of open surgery being the primary cause [24]. The pooled and subgroup quantitative analyses revealed results that were consistent with those identified for the operative time. These results could be significantly impacted by the presence of more than half of the pooled population undergoing minimally invasive gastrectomy.

The quantitative analyses of overall complications and length of hospital stay also revealed no significant differences between the two groups, aligning with the results of the individual studies included. It is crucial to emphasize that the DSF rates in the included studies were extremely low [16,19,21,22,30,31], in line with the findings of the literature. The most frequently reported complications were anastomotic fistulas [16,19,30], pancreatic fistulas/pancreatitis [16,19,21,30], anastomotic bleeding [16,30], intra-abdominal abscesses [19,21,24,30], intra-abdominal bleeding [16,24,30], wound infections [21,30], intestinal obstruction [21,30], and pneumonia [30]. The simultaneous presence of two or more complications made it impossible to determine the actual impact of the DSF and the duodenal stump reinforcement on the overall complication rate. None of the authors of the six included studies offered any hypothesis regarding the absence of significant differences between the two groups [16,19,21,22,30,31]. This also applies to major complications. Despite the significantly lower incidence of major postoperative complications in the reinforcement group compared to the control group (OR: 0.66, 95% CI: 0.43, 0.99, *p* = 0.04), the result should be interpreted with caution. This latter result could be closely related to the significantly lower rate of DSF in the reinforcement group compared to the control group. However, it is necessary to keep in mind the possibility that this finding was more or less markedly influenced by other contextual major postoperative complications after GC surgery.

Ultimately, the length of hospitalization did not reveal significant differences between the two groups, both in the individual studies included and in the pooled and subgroup quantitative analyses. None of the authors offered any speculation on this matter [16,19,21,22,30,31]. We believe that this outcome was significantly influenced by patient characteristics, the type of surgery performed, the presence and number of minor or major complications, the type of complication management, the hospital volume, the surgeon volume, and any enhanced recovery protocols adopted.

The present meta-analysis has several non-negligible limitations: (i) no RCTs were identified and none of the six observational studies presented a propensity-score matching analysis; (ii) the number of included studies was limited, despite the pooled population being large albeit markedly influenced by the size of the population of Sano et al.; (iii) DSF definitions not always available or different between studies; (iv) the heterogeneous nature of the populations encompassed all general demographic, surgical, pathological, and duodenal stump reinforcement characteristics. Finally, despite the fact that the publication bias test was not conducted in accordance with Cochrane’s methodological recommendations [32], it is impossible to rule out the potential for bias in one or more outcomes.

However, despite the above-mentioned limitations, it is necessary to underline the main strength of our manuscript. In the absence of RCTs and in the presence of many single-arm observational studies and few double-arm observational studies, our meta-analysis of just comparative studies represents the maximum statistical evidence on the topic of interest.

## 5. Conclusions

Our meta-analysis of comparative studies including patient populations undergoing distal/total gastrectomy for GC with or without duodenal stump reinforcement showed that the rate of DSF was significantly reduced in the reinforcement group compared to the control group. Furthermore, duodenal stump reinforcement appeared significantly correlated with the reduction in the rate of major postoperative complications.

The subgroup analyses, which focused on the chosen method of duodenal stump reinforcement and the laparoscopic group alone, appeared to support the findings obtained in the pooled population. Therefore, the most effective method for reducing the DSF rate seems to be the linear stapler transection of the duodenum, followed by reinforcement of the duodenal stump. However, the studies’ exclusive origin from Eastern countries and the significant heterogeneity of the populations included, in terms of patient characteristics, primary surgical pathology, surgery performed, and hospital and surgeon volumes, make it impossible to generalize and externally validate our results, especially in Western countries.

Therefore, our results need careful examination. Thus, well-designed randomized controlled trials or multicenter studies based on national and/or international registries are strongly needed, if we want to confirm our results.

## Figures and Tables

**Figure 1 cancers-17-01735-f001:**
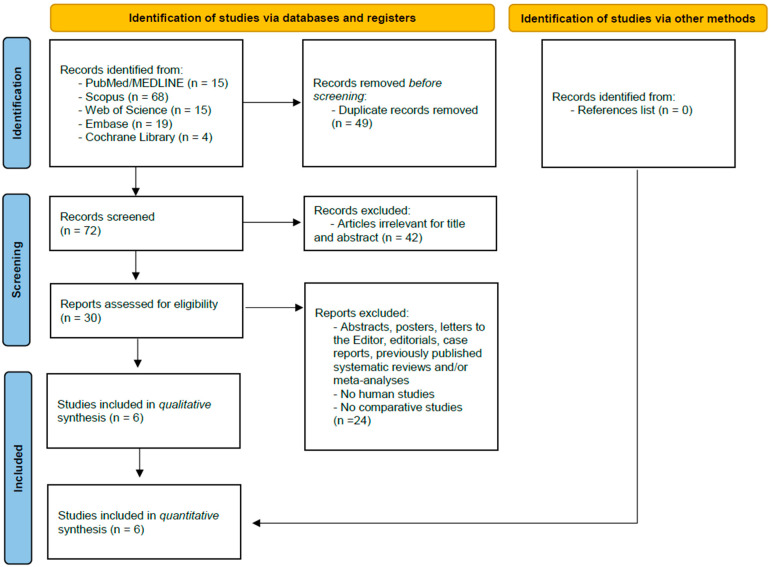
PRISMA flow chart of literature search.

**Figure 2 cancers-17-01735-f002:**
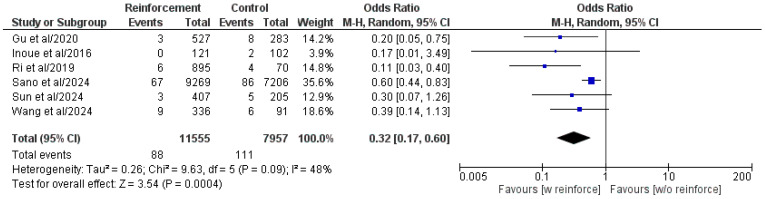
Forest plot comparing reported DSF rate between the Reinforcement and Control groups. CI, confidence interval; M-H, Mantel–Haenszel; w, with; w/o, without [16,19,21,22,30,31].

**Figure 3 cancers-17-01735-f003:**
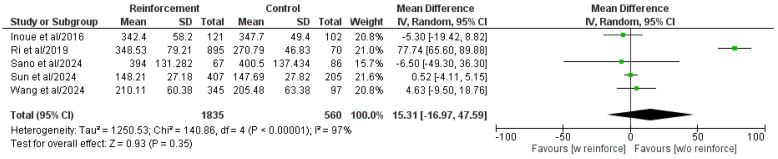
Forest plot comparing operative time between the Reinforcement and Control groups. SD, standard deviation; IV, inverse variance; CI, confidence interval; w, with; w/o, without [16,19,21,22,31].

**Figure 4 cancers-17-01735-f004:**
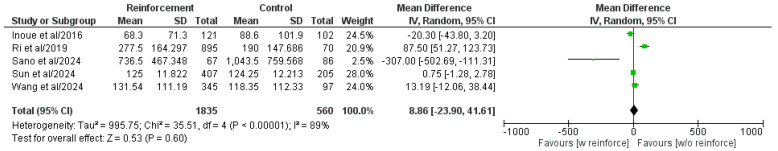
Forest plot comparing estimated blood loss between the Reinforcement and Control groups. SD, standard deviation; IV, inverse variance; CI, confidence interval; w, with; w/o, without [16,19,21,22,31].

**Figure 5 cancers-17-01735-f005:**
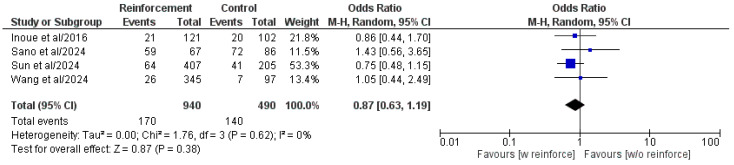
Forest plot comparing reported overall postoperative complications rate between the Reinforcement and Control groups. CI, confidence interval; M-H, Mantel–Haenszel; w, with; w/o, without [16,21,22,31].

**Figure 6 cancers-17-01735-f006:**
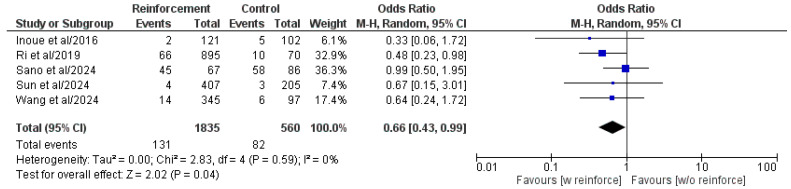
Forest plot comparing reported major postoperative complications (CD ≥ III) rate between the Reinforcement and Control groups. CI, confidence interval; M-H, Mantel–Haenszel; w, with; w/o, without [16,19,21,22,31].

**Figure 7 cancers-17-01735-f007:**
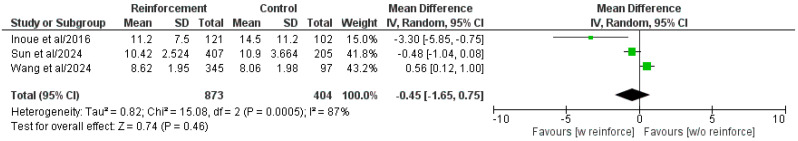
Forest plot comparing length of hospital stay between the Reinforcement and Control groups. SD, standard deviation; IV, inverse variance; CI, confidence interval; w, with; w/o, without [16,21,22].

**Table 1 cancers-17-01735-t001:** Study characteristics. *n* = number; DSF = duodenal stump fistula.

Authors/Year	Study Type		Study Centers, *n*	Study Country	Study Period	Patient Population, *n*	Duodenal Reinforcement, *n*		DSF, *n* (%)
							Yes	No	
Sano et al./2024 [31]	Retrospective	Multicenter	57	Japan	2012–2021	16,475	9269	7206	153 (0.93)
Wang et al./2024 [22]	Retrospective	Single-center	1	China	2022–2023	442	345	97	15 (3.39)
Sun et al./2024 [21]	Retrospective	Multicenter	2	China	2019–2023	612	407	205	8 (1.31)
Gu et al./2020 [30]	Retrospective	Multicenter	2	China	2013–2018	810	527	283	11 (1.36)
Ri et al./2019 [19]	Retrospective	Single-center	1	Japan	2005–2016	965	895	70	10 (1.04)
Inoue et al./2016 [16]	Retrospective	Single-center	1	Japan	2009–2014	223	102	121	2 (0.89)

**Table 2 cancers-17-01735-t002:** Population characteristics. *n* = number; DSF = duodenal stump fistula; SD = standard deviation; ASA = American Society of Anesthesiologists; CT = chemotherapy; RT = radiotherapy; n/a = not available.

Authors/Year	Duodenal Reinforcement	Patient Population, *n*	DSF, *n* (%)	Gender, *n*		Age (Years), Mean ± SD	BMI (kg/m^2^), Mean ± SD	ASA Score, *n*		Neoadjuvant CT ± RT, *n*	
				Male	Female			I–II	III–IV	Yes	No
Sano et al./2024 [31]	No	7206	86 (1.19)	11,729	4746	n/a	n/a	n/a	n/a	1538	14,937
	Yes	9269	67 (0.72)			n/a	n/a	n/a	n/a		
Wang et al./2024 [22]	No	97	6 (6.19)	71	26	60.85 ± 8.22	22.93 ± 3.07	70	27	n/a	
	Yes	345	9 (2.61)	243	102	60.05 ± 10.0089	22.746 ± 3.7624	261	84	n/a	
Sun et al./2024 [21]	No	205	5 (2.44)	128	253	67.90 ± 10.2	24.325 ± 0.674	174	31	n/a	
	Yes	407	3 (0.74)	77	154	69.22 ± 9.268	24.525 ± 0.659	322	85	n/a	
Gu et al./2020 [30]	No	283	8 (2.83)	596	214	62.5 ± 12.9	n/a	761	49	19	791
	Yes	527	3 (0.57)				n/a				
Ri et al./2019 [19]	No	70	4 (5.71)	60	10	63 ± 9.283	24.375 ± 3.101	895	0	n/a	
	Yes	895	6 (0.67)	626	269	61.5 ± 10.014	24.85 ± 3.849	70	0	n/a	
Inoue et al./2016 [16]	No	102	2 (1.96)	75	27	n/a	22.9 ± 3.2	99	3	n/a	
	Yes	121	0	79	42	n/a	23.4 ± 3.7	109	12	n/a	

**Table 3 cancers-17-01735-t003:** Surgical and histopathological characteristics. *n* = number; n/a = not available; Y = yes; N = no.

Authors/Year	Duodenal Reinforcement	Patient Population, *n*	Surgical Approach, *n*			Gastrectomy, *n*		Lymph Node Dissection, *n*		Drain		pT, *n*				pN, *n*		Stage, *n*			
			Open	Lap	Rob	Distal	Total	D1+	D2	Y/N	*n*	1	2	3	4	0	≥1	I	II	III	IV
Sano et al./2024 [31]	No	7206	9046	6556	873	7884	8591	7518	8957	n/a		6247	1759	4213	4256	8362	8113	n/a			
	Yes	9269																n/a			
Wang et al./2024 [22]	No	97	0	97	0	89	8	n/a		n/a		21	19	19	38	48	49	n/a			
	Yes	345	0	345	0	294	51	n/a				103	56	89	97	160	185	n/a			
Sun et al./2024 [21]	No	205	0	205	0	75	122	n/a		n/a		n/a				n/a		18	55	108	24
	Yes	407	0	407	0	130	285	n/a				n/a				n/a		45	91	236	35
Gu et al./2020 [30]	No	283	0	283	0	480	330	n/a		Y	1 or 2	220	98	151	341	353	457	261	172	377	0
	Yes	527	0	527	0			n/a													
Ri et al./2019 [19]	No	70	0	70	0	32	38	62	8	n/a		n/a				n/a		n/a			
	Yes	895	0	895	0	725	170	731	164			n/a				n/a		n/a			
Inoue et al./2016 [16]	No	102	0	102	0	102	0	70	32	Y	1	n/a				n/a		96	6	0	0
	Yes	121	0	121	0	121	0	82	39			n/a				n/a		106	10	5	0

**Table 4 cancers-17-01735-t004:** Duodenal stump reinforcement characteristics. USP = United States Pharmacopeia; n/a = not available.

Authors/Year	Duodenal Transection			Reinforcement Method	Suture Thread	
	Stapler	Cartridge Length, mm	Cartridge Closure Height, mm		Absorbable/Non-Absorbable	USP
Sano et al./2024 [31]	Linear Stapler	n/a	n/a	Unspecified suture	n/a	n/a
				Reinforced stapler	none	none
Wang et al./2024 [22]	Linear Stapler	n/a	n/a	Continuous suture	n/a	n/a
				Double half purse-string suture plus “8” pattern of stitching		
Sun et al./2024 [21]	Linear Stapler	60	3.5	Interrupted suture	Absorbable	3-0
				Purse-string suture		
Gu et al./2020 [30]	Linear Stapler	n/a	n/a	Interrupted suture	n/a	n/a
				Continuous suture		
				Semi-pouch suture		
				Complete-pouch suture		
Ri et al./2019 [19]	Linear Stapler	n/a	n/a	Interrupted suture	n/a	n/a
Inoue et al./2016 [16]	Linear Stapler	60	2.5	Interrupted suture	Absorbable	3-0

## Data Availability

The data presented in this study are available on request from the corresponding author.

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
