# Peer review of "Impact of Duodenal Stump Reinforcement in Preventing Duodenal Stump Fistula/Leakage After Distal or Total Gastrectomy for Malignant Disease: A Meta-Analysis of Comparative Studies"

_cancers, 2025, doi:10.3390/cancers17111735_

Round 1
Reviewer 1 Report
Comments and Suggestions for Authors
File attached

English language must be improved.
Author Response
see attached Word file.

Reviewer 2 Report
Comments and Suggestions for Authors
The manuscript is very well explained with a high level data and methodology regarding the research method. I have not detected any error regarding applied tests, statistics or results.
The paper's novelty should be outlined, as no plagism was found.
Even with no RCTs in the meta-analysis, since apparently there were not so far, the research strategy is fantastic.
Author Response
see attached Word file.

Reviewer 3 Report
Comments and Suggestions for Authors
Thank you for allowing me to review this first meta-analysis evaluating the impact of strengthening the duodenal stump suture after carcinological gastrectomy. The methodology is rigorous; objectives are clearly stated and the manuscript is well written and illustrated.
Neo-adjuvant treatment is currently recommended for locally developed gastric cancers. What was the impact of this treatment on the risk of duodenal fistula with and without reinforcement of the suture of the duodenal stump?
What classification of severe morbidity was used in this meta-analysis?
how to explain the lack of impact of duodenal fistula on hospitalization?
Did the authors analyze in their study the impact of duodenal fistula on survival without recurrence, overall survival and delay to completion of adjuvant chemotherapy when indicated.
Finally, I would add that all the series included are derived from the European Community and that the results may not be applicable to other continents.
Author Response
see attached Word file.

Reviewer 4 Report
Comments and Suggestions for Authors
The authors of the paper “Impact of duodenal stump reinforcement in preventing duodenal stump fistula/leakage after distal or total gastrectomy for malignant disease: A meta-analysis of comparative studies” aim to provide updated evidence by comparing the duodenal stump fistula (DSF) rates among patients who underwent distal or total gastrectomy for malignant gastric disease with or without reinforcement of duodenal stump.
They conclude that the duodenal stump reinforcement appears to reduce the rate of DSF after distal or total gastrectomy for malignant gastric disease.
This paper is interesting but there are several issues which should be defined in the analysis of the data or at least considered in the DISCUSSION.
MAJOR COMMENTS
- There is no definition of DSF. As a matter of fact, I do not know any formal definition of DSF, however I guess that the single authors have provided some definition of DSF in their original papers. Do they put together small, almost clinically inadvertent duodenal leakages, which sometimes would have been clinically silent unless discovered through the dosage of biliopancreatic enzymes, and massive output fluid?
- This is especially intriguing since the duration of the postoperative stay seems unaffected by the presence of a DSF. Do the authors have an explanation for this?
- How was the distribution of the drains in studies with different outcomes depending on the presence of reinforcement or not? If some patients were without drains, only symptomatic DSF can be diagnosed.
- The studies by Inohue, by Sun and by Wang did not find any impact of the reinforcement on the the occurrence of DSF. Did the authors of this manuscript speculate if the clinical variables of the patients of the above-mentioned studies were somewhat different from those which found a benefit from the reinforcement?
OTHER COMMENTS
- The tables contain many variables which are not considered in the analysis and weigh the reading down.
- The Discussion excessively relies on data of the literature rather than on the findings of their analysis
Author Response
see attached Word file.

Round 2
Reviewer 3 Report
Comments and Suggestions for Authors
the authors answered questions and comments on a point-by-point basis to improve the quality of the manuscript
Reviewer 4 Report
Comments and Suggestions for Authors
No comment